# Molecular Epidemiology of Group B Streptococci in Lithuania Identifies Multi-Drug Resistant Clones and Sporadic ST1 Serotypes Ia and Ib

**DOI:** 10.3390/pathogens11091060

**Published:** 2022-09-17

**Authors:** Jonah Rodgus, Ruta Prakapaite, Panagiotis Mitsidis, Ramune Grigaleviciute, Rita Planciuniene, Povilas Kavaliauskas, Elita Jauneikaite

**Affiliations:** 1Department of Infectious Disease Epidemiology, School of Public Health, Imperial College London, London W2 1PG, UK; 2MRC Centre for Molecular Bacteriology and Infection, Department of Infectious Disease, Imperial College London, London SW7 2AZ, UK; 3NIHR Health Protection Research Unit in Healthcare Associated Infections and Antimicrobial Resistance, Department of Infectious Disease, Imperial College London, Hammersmith Hospital, London W12 0NN, UK; 4Institute of Infectious Diseases and Pathogenic Microbiology, 59116 Prienai, Lithuania; 5Biozentrum, University of Basel, 4056 Basel, Switzerland; 6Biological Research Centre, Lithuanian University of Health Sciences, 47181 Kaunas, Lithuania; 7Institute of Microbiology and Virology, Lithuanian University of Health Sciences, 47181 Kaunas, Lithuania; 8Division of Infectious Diseases, Department of Medicine, Weill Cornell Medicine of Cornell University, New York, NY 10065, USA; 9Department of Microbiology and Immunology, University of Maryland School of Medicine, Baltimore, MD 21201, USA

**Keywords:** Group B Streptococcus, antimicrobial resistance, whole genome sequencing, MLST, mobile genetic elements

## Abstract

*Streptococcus agalactiae* (Group B Streptococcus, GBS) is a leading cause of neonatal infections. Yet, detailed assessment of the genotypic and phenotypic factors associated with GBS carriage, mother-to-baby transmission, and GBS infection in neonates and adults is lacking. Understanding the distribution of GBS genotypes, including the predominance of different serotypes, antimicrobial resistance (AMR) genes, and virulence factors, is likely to help to prevent GBS diseases, as well as inform estimates of the efficacy of future GBS vaccines. To this end, we set out to characterise GBS isolates collected from pregnant and non-pregnant women in Kaunas region in Lithuania. Whole genome sequences of 42 GBS isolates were analysed to determine multi-locus sequence typing (MLST), the presence of acquired AMR and surface protein genes, and the phylogenetic relatedness of isolates. We identified serotypes Ia (42.9%, 18/42), III (33.3%, 14/42), V (21.4%, 9/42), and a single isolate of serotype Ib. Genomic analyses revealed high diversity among the isolates, with 18 sequence types (STs) identified, including three novel STs. 85.7% (36/42) of isolates carried at least one AMR gene: *tetM* or *tetO* (35/42), *ermB* or *lsaC* (8/42) and a*nt6-Ia* and a*ph3-III* (2/42). This study represents the first genomic analysis of GBS isolated from women in Lithuania and contributes to an improved understanding of the global spread of GBS genotypes and phenotypes, laying the foundations for future GBS surveillance in Lithuania.

## 1. Introduction

Group B Streptococcus (GBS) is a *β*-haemolytic gram-positive bacterium associated with the colonisation of mucous membranes in the human body. A commensal in the gastrointestinal and lower rectogenital tracts of up to 36% of pregnant women in Europe [1,2], GBS can be transmitted from mother to neonate, with invasive GBS infections a leading cause of neonatal pneumonia, septicemia, and meningitis worldwide [3]. At present, the administration of intrapartum antimicrobials represents the only clinical intervention available to prevent GBS diseases in neonates [4]. However, antimicrobial resistance (AMR) and multi-drug resistance (MDR) conveyed by the presence of increasingly common genomic determinants (both genes and mutations) undermine the future viability of this intervention [5]. The presence of tetracycline resistance genes (*tetM* and *tetO*), for instance, is almost ubiquitous in GBS, with over 85% of isolates carrying either *tetM* or *tetO* [6,7]. Moreover, recent studies on GBS have reported an increase in macrolide, fluoroquinolone, and aminoglycoside resistance, driven by the genes *ermB*, *lnuC*, *mefA*, and *msrD* [5], mutations in *gyrA* and *parC*, and the genes *aac*(6’)-*aph*(2”), *ant6-Ia*, and *aph3-III* [5,6], respectively. Mutations in penicillin-binding-proteins (PBP), namely PBP1a, PBP2a, PBP2b, and PBP2x, that reduce the susceptibility of GBS to penicillin have been reported [8,9,10,11]. Mutations in *pbp1a* and *pbp2x*, specifically, have also been associated with reduced susceptibility to beta-lactams [12,13]. Given the increasing presence of these AMR determinants, there is a need for routine surveillance of GBS isolates and to better characterise circulating antimicrobial resistant GBS genotypes and phenotypes.

Monitoring the incidence of GBS serotypes, AMR determinants, and disease-causing GBS populations is likely to inform estimates of GBS vaccine efficacy. Of the ten known GBS serotypes (Ia, Ib, II-IX), just five serotypes, Ia, Ib, II, III, and V, account for 93-99% of cases of neonatal and adult infections globally [14]. As such, the main GBS vaccines in development include a trivalent vaccine covering serotypes Ia, Ib, and III [15]; a hexavalent vaccine covering serotypes Ia, Ib, and II-V [16,17], and a multivalent adjuvanted protein vaccine (NCT03807245). The latter of these vaccines exploits two fusion proteins constituting different combinations of two of the four most common Alp-family proteins, Alpha-C (also known as bca) and Rib, or Alp1 and Alp2/3 (https://www.minervax.com/product-pipeline/ accessed on 9 September 2022). Phase I clinical trials have shown high levels of antibody response in study participants, though further work is needed to investigate long term immunity and the potential interaction with pre-existing immunity to GBS [18,19].

Despite recognition of the need for a GBS vaccine more widely, there exists a distinct lack of detailed studies focused on the epidemiology of GBS carriage and infections in pregnant women, their neonates, and non-pregnant adults in Lithuania. Previous studies of GBS carriage in Lithuania have reported maternal and neonatal GBS carriage rates between 2006–2007 of 15.3% and 6.4% [20], respectively, with increased maternal GBS colonisation rates reported in later years (18.6% in 2012 and 22.9% in 2014 [21]), and mortality in neonates due to GBS infection having increased from 5.2% in 1999 to 18.1% in 2012 [22]. Here, we report the serotypes and genotypes of clones found among 42 clinical GBS isolates from pregnant and non-pregnant women in Lithuania, with findings laying the foundations for future surveillance studies in this country.

## 2. Results

### 2.1. Overview and Genomic Characterisation of Lithuanian GBS Isolates

Of the 42 GBS isolates, 83% (35/42) were retrieved from vaginal swab samples, 14% (6/42) from urine samples, and one from a blood sample of an individual with confirmed bacteraemia (Appendix A). Most isolates (85.7%, 36/42) were from women reported to be pregnant at the time of sample collection, with 32y the median age for this cohort (Appendix A). The isolates comprised four serotypes, Ia, Ib, III, and V. Serotype Ia was most prevalent, representing 42.9% (18/42) of isolates (including the single blood isolate, Appendix A). Serotypes III, V, and Ib represented 33.3% (14/42), 21.4% (9/42), and 2.4% (1/42) of isolates, respectively. We identified 18 distinct sequence types (STs) using multi-locus sequence typing (MLST), including three novel STs (ST1865-1867). The most common STs were ST17, ST1, ST88, and ST144 (Appendix A). STs clustered into five major clonal complexes (CCs), CC23 (n = 14), CC17 (n = 9), CC1 (n = 7), CC19 (n = 5), and CC452 (n = 4), each of which broadly corresponded to a major clade from the phylogenetic tree constructed for all isolates (Figure 1). 

We identified one serotype Ib ST1 isolate and two serotype Ia isolates from CC1 - one ST1 and one ST1387 (a single locus variant of ST1). As such, further investigations were carried out to identify any potential capsular switching events amongst the CC1 isolates (constituting mainly ST1 genotypes). Compared to other publicly available serotype Ia ST1 isolates (n = 8), our CC1 isolates of serotypes Ia (n = 2), V (n = 4), and Ib (n = 1) appeared to have participated in a capsular switching event. Indeed, all serotype Ia isolates, except two (LTS018 and SRR7283561), were genetically similar to other serotype V isolates, as well as the serotype Ib ST1 isolate, with less than 90SNPs (recombination-free core SNPs) present (Appendix A). Moreover, recombination borders were very similar for the Ib isolate (LTS066) and two Ia isolates (LTS045 and LTS018), strongly indicative of a capsular switching event between serotypes V and Ib, and serotypes V and Ia (Appendix A). 

### 2.2. Antimicrobial Resistance Profiles and Antimicrobial Resistance Associated Mobile Genetic Elements

Overall, 64% (27/42) of isolates carried only one tetracycline resistance gene, either *tetM* (n = 25) or *tetO* (n = 2). Two isolates (LTS011 and LTS031) carried one tetracycline resistance gene (*tetM* or *tetO*), the erythromycin resistance gene, *ermB*, as well as the aminoglycoside resistance genes, *ant6-Ia* and *aph3-III* (Figure 1). Nine isolates carried the macrolide resistance genes *ermA*, *ermB*, and *lsaC* (Figure 1). Four of the five isolates carrying *ermB* had confirmed resistance to erythromycin and clindamycin (Figure 1, Appendix A). Six serotype Ia ST88 isolates did not carry any acquired AMR genes. All isolates were susceptible to penicillin and, as expected, did not carry any mutations in the genes *pbp1A* or *pbp2X*, previously associated with resistance to penicillin (Appendix A). Additionally, we identified no mutations attributed to fluoroquinolone resistance in the quinolone resistance-determining regions, *gyrA* and *parC* [23]. 

Acquired AMR genes are often carried by mobile genetic elements (MGEs). Hence, we set out to investigate the presence of MGEs in our GBS isolates. The most common transposon, Tn*916*, known to carry *tetM*, was detected in 14 isolates. Furthermore, Tn*3872* (Appendix A), carrying *tetM* and *ermB*, was detected in two ST1 isolates (LTS039 and LTS066). In one of the two multi-drug resistant isolates (LTS011, serotype III, ST17), we detected a putative integrative conjugative element (ICE, named here as LTS_ICE*Sag37*), downstream of the recombination hotspot gene, *rplL*, with this ICE carrying the acquired AMR genes, *tetO*, *ant6-Ia*, *aph3-III*, and *ermB*. Reconstruction of LTS_ICE*Sag37* showed that, at 120Kbp, it was much longer than the 79Kbp ICE*Sag37* reference sequence described previously [24]. This was due to the potential insertion of a region containing the phage assembly and recombination genes present in the *Sag153* reference genome. However, LTS_ICE*Sag37* was split across three contigs and therefore requires long-read sequencing to confirm such an insertion.

### 2.3. Distribution of Major Surface Proteins

Most notably, a surface protein required for hypervirulence, HvgA, was present in the genomes of all serotype III ST17 isolates (Figure 1). All except one isolate carried genes for one of the four proteins, *alpha-C*, *alp1*, *alp2/3*, or *rib* (Table 1), targeted by the protein-based GBS vaccine currently in trial. All serotype Ia isolates (n = 18) and most of the serotype V isolates (n = 8/9) carried a serine-rich repeat glycoprotein determinant, *ssr1*, while *ssr2* was identified only in serotype III ST17 isolates that were also positive for *hvgA* (Table 1). 66% (28/42) of isolates carried two of the pilus genes, while 29% (12/42) of isolates carried *PI-2A* only (Table 1).

## 3. Discussion

This study represents the first in-depth genomic characterisation of GBS isolates from Lithuania. Here, we have identified four known serotypes, with serotypes Ia, III, and V predominating in agreement with studies of GBS carriage and disease in pregnant women [14], and of GBS carriage in pregnant women and neonates in Lithuania [20]. The isolates in our dataset showed high genetic diversity, covering 18 sequence types based on MLST profiles. These sequence types clustered into seven clonal complexes, of which CC23, CC17, and CC1 were most common, corroborating findings from previous studies of GBS carriage [7,14]. Notably, one serotype Ib ST1 isolate carried the macrolide resistance gene, *ermB*, with similar isolates having been characterised in Portugal as an emergent macrolide resistant lineage [25]. Furthermore, we identified one serotype Ia ST1 isolate and one serotype Ia ST1387 isolate belonging to CC1, indicating some serotype Ia isolates may have participated in a capsular switching event similar to other recombination events reported in the ST1 lineage [26]. However, none of the serotype Ia CC1 isolates in our dataset carried macrolide resistance genes, nor was serotype Ia the most common serotype belonging to CC1, with this cluster consisting primarily of serotype V isolates, as may have been anticipated [7,25]. 

Following local guidelines, antimicrobial susceptibility testing was not requested for every isolate. Of those tested, all showed susceptibility to benzylpenicillin, with no mutations in *pbp1a* or *pbp2X* detected, supporting the general assumption that GBS’s susceptibility to this antibiotic is high, despite studies having reported varying degrees of penicillin resistance in GBS [11,27]. Nine GBS isolates carried genes conveying resistance to erythromycin and clindamycin, in line with reports from other European countries including Serbia [28], Portugal [25], and the UK [29]. In one serotype III ST17 isolate, we detected a multi-drug resistant MGE carrying tetracycline (*tetO*), macrolide (*ermB*), and aminoglycoside (*aph3-III* and *ant6-Ia*) resistance genes. This MGE had a backbone of the well described ICE, ICE*Sag37* [24], which has been previously identified in multi-drug resistant serotype III ST17 isolates [30]. That we identified a MGE carrying AMR and MDR determinants in our isolates underscores the capacity of GBS populations to exchange genetic material facilitating their adaptation to the selection pressures imposed by antimicrobials. This phenomenon has been described in other streptococcal species, such as *Streptococcus pneumoniae* [31]. Therefore, further studies are needed to track the genomic changes occurring in GBS populations.

Currently, two strong GBS vaccine candidates are being considered for implementation in vaccination programs for pregnant women to prevent infections in their neonates. One of these is a hexavalent polysaccharide vaccine, covering serotypes Ia, Ib, II, III, IV, and V [16,17], and the other is a multivalent adjuvanted protein vaccine (NCT03807245). Our findings suggest the hexavalent vaccine would offer full coverage in Lithuania, supporting findings from a recent review that reported 93–99% of cases of neonatal and adult infections would be covered by this vaccine [14]. A Minervax protein-based GBS vaccine is also in phase II clinical trials (https://www.minervax.com/product-pipeline/ accessed on 9 September 2022). As this vaccine exploits the N-terminal domains of the most prevalent alpha-like surface proteins (Alpha-C (bca), Rib, Alp1, and Alp2/3), and 97.6% (41/42) of our GBS isolates carried genes for one of these four proteins, it should offer a similarly high level of coverage in Lithuania. However, alpha-like surface proteins were detected here using only in silico genotyping methods that do not necessarily predict the expression of these proteins in isolates.

Unfortunately, too few isolates were investigated in this study to draw any consequential conclusions regarding the overall prevalence of specific AMR profiles and MGEs in the GBS population in Lithuania. Indeed, the serotypes and genotypes presented here may not represent the national picture of disease-causing GBS in Lithuania. A second limitation of this study relates to the ability to fully characterise MGEs using Illumina short reads. In future work, long-read sequencing technologies such as Nanopore sequencing, which can generate ultra-long (>4Mb) reads, may be implemented to better reconstruct and characterise the MGEs present in GBS isolates. Nonetheless, this study contributes a valuable first perspective on the MGEs present in the GBS population in Lithuania.

Taken together, our findings lay the foundations for future GBS surveillance studies in Lithuania, enabling future genomic and epidemiological comparisons to other GBS isolates that may inform national antimicrobial procedures to prevent GBS diseases in the same region. All such work is vital to better understanding the global spread of AMR among GBS and estimating the efficacy of GBS vaccines once available. 

## 4. Materials and Methods

### 4.1. Bacterial Isolates, Microbiological and Demographic Data

Isolates were collected between July 2020–September 2020 from participating diagnostic laboratories in Kaunas region (Lithuania) as a part of Surveillance and Molecular Epidemiology of Maternal Infections Consortium. In total, 35 isolates were retrieved from vaginal swab samples, six from urine samples and one from a blood sample of an individual with confirmed bacteraemia (Appendix A). Vaginal swabs, urine and blood samples were processed following Clinical Laboratory Standards Institute (CLSI) guidelines and local microbiology diagnostic procedures at the participating hospitals. GBS was identified by distinct β-haemolysis on Columbia agar with 5% Sheep Blood (Becton Dickinson GmbH, Heidelberg, Germany) and further confirmed by latex agglutination with a Streptococcal grouping kit (Oxoid, Dublin, Ireland). The following demographic information was collected (where available): age of patient, pregnancy status, sample type from which GBS was isolated, date of GBS isolation and results from routine antibiotic susceptibility testing. Antimicrobial susceptibility testing for penicillin, erythromycin, clindamycin and sulfadiazine-trimethoprim was performed using a Kirby-Bauer disk diffusion assays following the standard protocols [32] and results were interpreted using the EUCAST breakpoint table v.10 (https://www.eucast.org/fileadmin/src/media/PDFs/EUCAST_files/Breakpoint_tables/v_10.0_Breakpoint_Tables.pdf accessed on 1 November 2020). Phenotypic antibiotic susceptibility testing was carried out following local microbiology diagnostic procedures. Only some isolates were tested against all four antibiotics. Following the local Lithuanian screening routines, all pregnant females are routinely screened for Group B Streptococcus colonization between 35th and 37th week of pregnancy. In the absence of symptoms or risk factors, the attending physician, or the patients themselves, have the right to omit any further antimicrobial treatment at the time of testing and therefore the antimicrobial testing can be not requested by the physician. This is, however, subject to change in case new risk factors or symptoms emerge.

### 4.2. DNA Extraction and Whole Genome Sequencing 

GBS isolates were streaked on Columbia Blood agar plates (Oxoid, Basingstoke, UK) and incubated at 37 °C 5% CO_2_ overnight. Genomic DNA from GBS bacterial isolates was extracted using the GenElute^TM^ bacterial Genomic DNA kit (Sigma-Aldrich, Burlington, MA, USA) following the manufacturer’s instructions for gram-positive bacteria with modifications as follows: GBS was lysed in 180 μL G+ lysis solution (comes with the GenElute^TM^ bacterial Genomic DNA kit) with added 20 μL mutanolysin (Sigma-Aldrich, US; prepared at 3000 U/mL) and 20 μL lysozyme (Sigma-Aldrich, USA; prepared at 100 mg/mL) prior to incubation at 37 °C for 1 h. Subsequent steps followed the manufacturer’s instructions. DNA was quantified using a NanoDrop spectrophotometer (Thermo, Waltham, MA, USA). 

Multiplexed DNA libraries were prepared using Nextera XT DNA Library Preparation kit (Illumina, San Diego, CA, USA) following manufacturer’s instructions and whole genome sequenced on Illumina HiSeq 2500 System (Illumina) using 2 × 150 bp paired-end mode. Read demultiplexing and adapter trimming were performed by the Imperial BRC Genomics Facility using their production pipeline.

### 4.3. Genomic Analyses 

Quality checks. Raw sequencing reads were checked with FastQC v. 0.11.9 (https://www.bioinformatics.babraham.ac.uk/projects/fastqc/) before raw reads were trimmed with Trimmomatic v.0.39 [33] using the following parameters: remove trailing low quality or N bases (below quality three); scan the read with a 5-base wide sliding window, cutting when the average quality per base drops below 30 and drop reads below 50 bases long. Bacterial species from assembled genomes was confirmed using Kraken2 v2.0.8-beta and its full bacterial database [34]. 

De novo assembly and annotation. Unicycler 0.4.8 [35] was used to generate consensus de novo assemblies. Assembly statistics were checked using QUAST v5.0.2 [36] and assembly graphs were visualised using Bandage [37]. Assemblies were annotated using Prokka v.1.13.4 [38].

Serotyping and multi-locus sequence typing (MLST). In-silico serotyping was done by iPCRess v2.2 (https://www.ebi.ac.uk/about/vertebrate-genomics/software/ipcress-manual) using GBS multiplex primers as reported by Imperi et al., 2010 [39]. MLST was assigned using mlst (Seemann T, mlst Github https://github.com/tseemann/mlst and the PubMLST database (https://pubmlst.org/). Novel allele sequences and allelic combinations were submitted to the PubMLST database (https://pubmlst.org/). Clonal complexes were assigned using the PubMLST database (https://pubmlst.org/).

Characterisation of surface proteins. Blastn [40] was used to screen for surface protein genes: *alpha-C*, *alp1*, *alp2/3*, *rib*, *ssr1*, *ssr2* and *PI-1*, *PI-1B*, *PI-2A* (including all four allelic variants *PI-2A1*, *PI-2A2*, *PI-2A3* and *PI-2A4*), *PI-2B,* and *PI-2B2* using surface protein gene sequences described by McGee et al [7] as references. During Blastn analysis, sequence queries of ≥95% identity were reported as positive results, but we have also recorded surface protein matches with <95% identify, though we did not interpret these as positive results in this study.

Detection of antimicrobial resistance determinants. Acquired antimicrobial resistance genes were detected from assemblies using ABRicate (Seemann T, Abricate, Github https://github.com/tseemann/abricate) with the ARG-ANNOT [41] and ResFinder [42] databases. Chromosomal genes: *gyrA, parC, pbp1a,* and *pbp2x,* were investigated using Clustal Omega [43] for the presence of mutations conferring resistance to fluoroquinolones and beta-lactams, as previously reported by McGee et al. [7].

Characterisation of AMR-carrying mobile genetic elements. Annotated genomes were visualised with ARTEMIS [44]. Classification of known mobile genetic elements (MGEs) was completed by manually scanning the regions proximal to acquired AMR genes and subsequently identifying characteristic genes of previously described integrative and conjugative elements (ICEs): insertion sequences (*IS*), transposases (*tnp*), integrases (*int*), excisionases (*xis*), relaxases (*mobC*), helicases (*recB*), and/or recombinases. Characteristic genes involved in conjugation included those of Type IV-secretion systems and the *vir* family. Regions were also scanned for *rumA* and *rplL* genes which are recombination hotspots for ICE integration. Sequences of potential MGEs were compared to a reference found in TnRegistry (https://transposon.lstmed.ac.uk/tn-registry). In genomes where no ICE characterising genes were identified, genes adjacent to acquired AMR genes were investigated manually using ARTEMIS and their annotations checked using BLASTn. Figures were created with EasyFig [45] and the built-in BLASTn alignment tool.

Phylogenetic analysis. Single nucleotide polymorphisms (SNPs) were detected by mapping trimmed raw reads to a reference genome SS1 (CP010867.1) using snippy v4.6.0 (https://github.com/tseemann/snippy). The whole genome alignment was cleaned using the snippy-clean_full_aln function available as part of Snippy. Gubbins v2.4.1 [46] was used to identify recombination regions before processing alignment with SNP-sites [47] to obtain a recombination-free core SNPs file. The maximum-likelihood phylogenetic tree was constructed using IQ-TREE v.2.0.3 [48] with the “GTR+G+ASC” model (which combines the following models: general time reversible model with unequal rates and unequal base frequency (GTR model); discrete Gamma model with default 4 rate categories (G model) and an ascertainment bias correction model (ASC)) and 1000 bootstrap replicates. The maximum-likelihood phylogenetic tree was visualised using iTOL v.5 [49]. SNP pairwise distances were calculated using MEGA X v.10.2.4 [50]. To check for potential capsular switching for ST1 serotype Ia isolate, all available ST1 isolates reported as serotype Ia (by genotype) on the PubMLST database (https://pubmlst.org/) were collected (Appendix A) and analysed in combination with serotype Ib ST1, serotype V ST1 and serotype Ia ST1387 isolates, constituting CC1. All isolates underwent the phylogenetic analysis described above. Recombination regions were visualised using Phandango [51] and BRIG [52].

## Figures and Tables

**Figure 1 pathogens-11-01060-f001:**
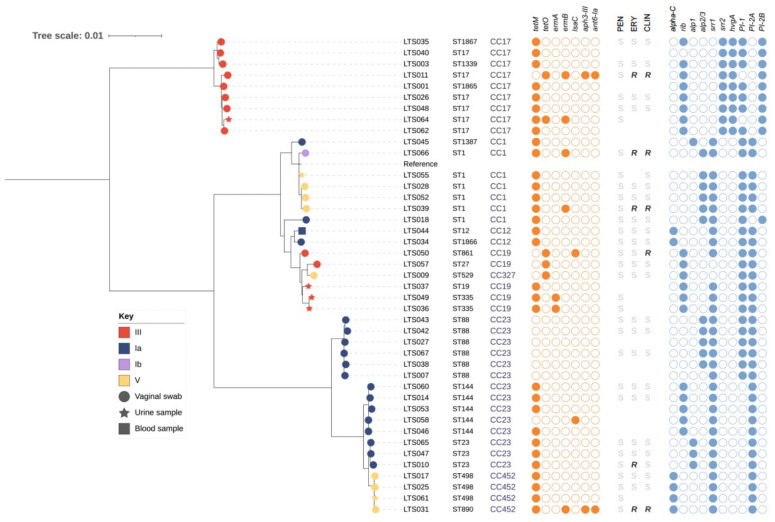
Maximum likelihood phylogenetic tree for GBS isolates from women in Lithuania. Maximum likelihood mid-point rooted phylogenetic tree constructed using recombination-free alignment of core-genome SNPs and the reference genome, SS1 (accession number: CP010867), representing serotype V ST1. The first data column shows the ST of each isolate per MLST results; the second data column shows the CC of each isolate. Branch colour shows the serotype of each isolate: Ia (blue), Ib (purple), III (red), and V (yellow). Branch shape shows the sample type of each isolate: vaginal (circle), urine (star), and blood (square). The next six data columns show the presence (filled circles) and absence (unfilled circles) in each isolate of the acquired AMR genes, *tetM*, *tetO*, *ermA*, *ermB*, *lsaC, aph3-III*, and *ant6-Ia*. The next three data columns show the susceptibility (S) or resistance (R) of the isolates to the antimicrobials, benzylpenicillin (PEN), erythromycin (ERY), and clindamycin (CLIN). Empty space in these columns indicates that the antibiotic susceptibility testing result was not required for treatment purposes and therefore not available. The final ten data columns show the presence (filled circles) and absence (unfilled circles) in each isolate of the surface proteins and virulence genes, *alpha-C* (or *bca*), *rib*, *alp1*, *alp2/3*, *srr1*, *srr2*, *hvgA*, *PI-1*, *PI-2A*, and *PI-2B*. Tree scale indicates nucleotide substitution rate per site.

**Table 1 pathogens-11-01060-t001:** Summary of the surface protein genes identified in the 42 Lithuanian GBS isolates.

GBS Serotype (n)	*alpha-C*	*alp1*	*alp2/3*	*rib*	*srr1*	*srr2*	*PI-1*	*PI-2A*	*PI-2B*	*hvgA*
Ia (n = 18)	2	4	7	5	18	0	10	17	1	0
Ib (n = 1)	0	0	1	0	1	0	1	1	0	0
III (n = 14)	0	0	0	13	4	9	12	5	9	9
V (n = 9)	4	0	4	1	8	0	5	9	0	0

## Data Availability

The whole genome raw sequence data and assemblies have been submitted to the European Nucleotide Archive (https://www.ebi.ac.uk/ena/), under the project accession PRJEB55405 (Appendix A).

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
