# Peer review of "Molecular Epidemiology of Group B Streptococci in Lithuania Identifies Multi-Drug Resistant Clones and Sporadic ST1 Serotypes Ia and Ib"

_pathogens, 2022, doi:10.3390/pathogens11091060_

Round 1

Reviewer 1 Report

In this study, the authors performed whole genome sequencing using short-read technology on 42 clinical Group B Streptococcus isolates from pregnant and non-pregnant women in Lithuania. The genomes were analysed and the authors have provided a report on the phylogeny of the genomes in addition to target-specific in silico analysis including multilocus sequence typing, serotyping (based on genotyping), prevalence of antimicrobial resistance genes, and prevalence of genes encoding surface antigens.

MAJOR COMMENTS

·       The Introduction section is short – I would recommend the authors to add more info about the proteins present in the multivalent protein vaccine being developed against GBS. Given this paper is about isolates from Lithuania, I would have liked to see something about the prevalence of GBS disease and disease surveillance in Lithuania.

·       The statement on lines 82-83 must be in the Discussion section, not Results.

·       Gene names/symbols must be in italics and start with a small letter – correct in Figure 1, Table 1, line 105, line 108 and throughout Results section.

·       It is unclear why some antimicrobial resistance phenotypic data are missing in Figure 1. Please add this information or explain why such data were not available. I strongly believe the phenotypic data are important for this manuscript and would recommend the authors to test the isolates with missing phenotypic data.

·       Please show the clonal complex (CC) information on the phylogenetic tree in Figure 1.

·       The authors mention potential vaccine coverage in the third paragraph of the Discussion – a mention about the inability of genotyping methods to predict protein expression would have been nice to see.

MINOR COMMENTS

  1. Line 50: “Gram-positive” should read “gram-positive”. Small letter ‘g’ because of the hyphen.
  2. Line 57: “are conveyed” should read “have been reported in” and authors should cite at least one reference for this statement.
  3. Line 64: “infections” should read “GBS infections”
  4. Line 67: “vaccines” should read “vaccine”
  5. Line 70: “vaccines” should read “vaccine”
  6. Line 84: “ST” must be written in full the first time it appears in the manuscript (excluding the Abstract)
  7. Lines 84-86: There is repetition about discovery of the novel STs – remove.
  8. Line 86: “STs clustered” should read “The identified STs clustered”
  9. Line 90: “SLV” abbreviation used – state that it stands for “single locus variant”
  10. Line 92: “Comparison” should read “In comparison”
  11. Line 125: “we have looked” should read “we looked”
  12. Lines 149 to 151: revise the sentences – at least two prepositions are missing.
  13. Line 171” “had macrolide resistance genes present” should read “possessed macrolide resistance genes”
  14. Line 176: “no resistance conferring mutations” should read “no mutations conferring resistance”
  15. Line 181: revise the English
  16. Lines 245-246: It is unclear how the lysis buffer with mutanolysin and lysozyme was prepared. The authors mention a total volume of 40 mL (20 mL + 20 mL), but what volume of this mixture was used for each extraction?
  17. Line 246: “1 hours” should read “1 hour”
  18. Line 268: “pubMLST database” should read “the PubMLST database” – correct on lines 270, 271 and 305 as well.
  19. Line 279: “interrogated” should read “analysed” or “investigated”
  20. Line 286: “transposes” should read “transposases”
  21. Line 286: “Tnp, Int, Xis, MobC and RecB” should not be italicised if referring to the proteins. But if referring to genes, then the first letter should be a small letter.
  22. Line 287: “recombinases” should not be in italics

Author Response

We thank the reviewer for their time to read our manuscript and make suggestions for improvements. We have taken time to edit the manuscript accordingly and we believe that the manuscript has been improved. 

Point-by-point responses to reviewer comments are as follows (page and line numbers indicated based on clean version of the updated manuscript): 

Reviewer #1

Comments and Suggestions for Authors

In this study, the authors performed whole genome sequencing using short-read technology on 42 clinical Group B Streptococcus isolates from pregnant and non-pregnant women in Lithuania. The genomes were analysed and the authors have provided a report on the phylogeny of the genomes in addition to target-specific in silico analysis including multilocus sequence typing, serotyping (based on genotyping), prevalence of antimicrobial resistance genes, and prevalence of genes encoding surface antigens.

MAJOR COMMENTS

  • The Introduction section is short – I would recommend the authors to add more info about the proteins present in the multivalent protein vaccine being developed against GBS. Given this paper is about isolates from Lithuania, I would have liked to see something about the prevalence of GBS disease and disease surveillance in Lithuania.

Authors’ response: We have now added the requested information on proteins that are present in the multivalent protein vaccine against GBS and added information on surveillance in Lithuania to the introduction, please see page 2, lines 59-90.

  • The statement on lines 82-83 must be in the Discussion section, not Results.

Authors’ response: We have moved this sentence from the results to the discussion section and have made overall improvements to the discussion section to reflect further suggestions from reviewers.

  • Gene names/symbols must be in italics and start with a small letter – correct in Figure 1, Table 1, line 105, line 108 and throughout Results section.

Authors’ response: Thank you to the reviewer for spotting that not all genes in the text were in italics. We have now double-checked these throughout the text and have put gene names in italics where appropriate. Pilus genes PI-x were left in capital letters as this is the common way to write these genes in GBS literature.

  • It is unclear why some antimicrobial resistance phenotypic data are missing in Figure 1. Please add this information or explain why such data were not available. I strongly believe the phenotypic data are important for this manuscript and would recommend the authors to test the isolates with missing phenotypic data.

Authors’ response: We agree that phenotypic antimicrobial susceptibility data is very useful. Unfortunately, for this study we were not able to re-test the antimicrobial susceptibility for the isolates–acquired as a part of prenatal pregnant female screening program–and relied on the data provided by the participating hospitals that carried out the antimicrobial susceptibility testing following their local microbiology diagnostics procedures. Following the local Lithuanian screening routines, all pregnant females are routinely screened for Group B Streptococci colonization between 35th and 37th week of pregnancy. In the absence of symptoms or risk factors, the attending physician–or the patient themselves–has the right to omit any further antimicrobial treatment at the time of testing, therefore the antimicrobial testing can be not requested by the physician. This is, however, subject to change in case new risk factors or symptoms emerge. All co-authors agreed that in this case, it was better providing the phenotypic information that we did have, rather than omitting it all together. We have added this information to the Materials and Methods section(page 6-7, lines 272-278) and clarified in the Figure 1 legend that antibiotic susceptibility testing results were not available for all isolates (page 4, lines 134-136).

  • Please show the clonal complex (CC) information on the phylogenetic tree in Figure 1.

Authors’ response: We have now included clonal complex information to Figure 1.

  • The authors mention potential vaccine coverage in the third paragraph of the Discussion – a mention about the inability of genotyping methods to predict protein expression would have been nice to see.

Authors’ response: We have added the requested information on GBS vaccine candidates in the discussion section, please see page 6, lines 220-233.

MINOR COMMENTS

  1. Line 50: “Gram-positive” should read “gram-positive”. Small letter ‘g’ because of the hyphen.

Authors’ response: done.

  1. Line 57: “are conveyed” should read “have been reported in” and authors should cite at least one reference for this statement.

Authors’ response: We have replaced the wording as suggested by the reviewer and have included the references to illustrate the examples of studies reporting increase in number of antimicrobial resistant and multidrug resistant GBS isolates.

  1. Line 64: “infections” should read “GBS infections”

Authors’ response: “GBS infections” has been added.

  1. Line 67: “vaccines” should read “vaccine”

Authors’ response: this has been corrected.

  1. Line 70: “vaccines” should read “vaccine”

Authors’ response: this has been corrected.

  1. Line 84: “ST” must be written in full the first time it appears in the manuscript (excluding the Abstract)

Authors’ response: this has now been added.

  1. Lines 84-86: There is repetition about discovery of the novel STs – remove.

Authors’ response: the repetition about discovery of the novel STs has been removed.

  1. Line 86: “STs clustered” should read “The identified STs clustered”

Authors’ response: the suggested addition of “The identified..” has been included at the beginning of the sentence.

  1. Line 90: “SLV” abbreviation used – state that it stands for “single locus variant”

Authors’ response: the SLV abbreviation has been now explained.

  1. Line 92: “Comparison” should read “In comparison”

Authors’ response: this has been now corrected.

  1. Line 125: “we have looked” should read “we looked”

Authors’ response: this has been corrected.

  1. Lines 149 to 151: revise the sentences – at least two prepositions are missing.

Authors’ response: this has now been corrected and both sentences have been rephrased.

  1. Line 171” “had macrolide resistance genes present” should read “possessed macrolide resistance genes”

Authors’ response: this has been corrected, line xx.

  1. Line 176: “no resistance conferring mutations” should read “no mutations conferring resistance”

Authors’ response: this has been corrected, line xx.

  1. Line 181: revise the English

Authors’ response: the first sentence in the discussion has been revised.

  1. Lines 245-246: It is unclear how the lysis buffer with mutanolysin and lysozyme was prepared. The authors mention a total volume of 40 mL (20 mL + 20 mL), but what volume of this mixture was used for each extraction?

Authors’ response: additional information on the volume of the G+ lysis solution (which is provided as part of the GenElute bacterial DNA extraction kit) has been added. As well as typo corrected to indicate that microliters not mililitters of mutanolysin and lysozyme were used. Concentration for mutanolysin and lysozyme that were used for the bacterial lysis are aslo indicated in the brackets. Please see page 7, lines 285-287.

  1. Line 246: “1 hours” should read “1 hour”

Authors’ response: this has been corrected.

  1. Line 268: “pubMLST database” should read “the PubMLST database” – correct on lines 270, 271 and 305 as well.

Authors’ response: this has now been corrected.

  1. Line 279: “interrogated” should read “analysed” or “investigated”

Authors’ response: “interrogated” has been replaced with “investigated”.

  1. Line 286: “transposes” should read “transposases”

Authors’ response: this has been corrected.

  1. Line 286: “Tnp, Int, Xis, MobC and RecB” should not be italicised if referring to the proteins. But if referring to genes, then the first letter should be a small letter.

Authors’ response: this has been corrected by editing the gene names to start with the smaller letter.

  1. Line 287: “recombinases” should not be in italics

Authors’ response: done.

Reviewer 2 Report

This paper, which aim was the broad molecular characterization of GBS isolates obtained from Lithuanian women, presents interesting and relevant knowledge. Furthermore, since there are no such a studies on GBS in the Lithuanian population, the results presented are crucial to “better understand global spread of GBS genotypes and antimicrobial resistance patterns” – as the authors summarized their paper. In turn, determination of GBS serotypes distribution may lead to the development of an innovative vaccine that would protect, above all, newborns against life-threatening GBS infections. The methods used in the presented study were appropriate, however some clarification should be considered. I recommend that paper to be accepted for publication due to the importance of the topic, but after a minor essential revision that addresses the following concern. My major comments are listed below.

 Introduction: Please briefly describe the genes, which were examined in your study (gyrA, parC, pbp1a, pbp2x)

Line 59: Please, rephrase the following expression: “clear need”, please rephrase “circulating antimicrobial resistant phenotypes”

Line 61: In the sentence: “The surveillance of circulating GBS serotypes and genotypes will provide essential information for estimating GBS vaccine efficacy in the future” there is too far-reaching cause-and-effect relationship, please improve the sentence

Line 77: What was the source of remaining isolates?

Line 81: Please verify the percentage values. My calculations indicate, that serotype III is present in 33,33% of examined isolates.

Lines 84-85: Information of three novel ST’s is duplicated. Please improve it.

Line 118: Please rephrase the following sentence: “Six isolates, all of them serotype Ia genotype ST88, did not carry any acquired AMR genes.”

Line 138: Please rephrase the following sentence: “Figure 2. Previously described novel multidrug resistance MGE ICESag37 found in GBS isolates from Lithuania.” I suggest to avoid “previously”

Line 140: use italics in the species name

Line 206: Please verify this sentence: “In our dataset 97.6% (41/42) of GBS isolates had at least one of these four proteins, suggesting that this vaccine could also offer high level of coverage for these isolates “. For my best knowledge only one Alp protein can be present in the individual bacterial strain, so the expression “at least one” may be misleading.

Line 208: Please revise: “Notwithstanding”

Line 234: Please revise: “Where available anonymised, double-coded data on clinical diagnosis, age of the patient and routine antibiotic susceptibility testing result was collected.”

Lines 235-237: Please name the examined antibiotics.

Line 245: Please include the name of the producers of mutanolysin and lysozyme

Line 227: section “Materials and Methods”, please briefly describe how the samples were collected. The following sentence: “Of the 42 GBS isolates, 35 isolates were retrieved from vaginal swab samples, six from 75 urine samples, and one from a blood sample of an individual with confirmed bacteraemia.” As well as “…with the median age of 32y for this cohort” should be moved to “Material and Methods” section. Was each sample collected from individual patient or some samples (e.g. swab and urine) had been collected from the same patient? How bacterial cultivation was carried out?

Lines 272-274: Please state what surface proteins were studied. More detailed procedures of the proteins’ detection would improve the paper value. Please improve.

Line 279: Double space, please improve

Line 291: Please rephrase “Where no characteristic genes were identified…”

Line 301: Please develop the phrase “GTR+G+ASC”

Author Response

We thank the reviewer for their time to read our manuscript and make suggestions for improvements. We have taken time to edit the manuscript accordingly and we believe that the manuscript has been improved. 

Point-by-point responses to reviewer comments are as follows (page and line numbers indicated based on clean version of the updated manuscript): 

Reviewer #2

Comments and Suggestions for Authors

This paper, which aim was the broad molecular characterization of GBS isolates obtained from Lithuanian women, presents interesting and relevant knowledge. Furthermore, since there are no such a studies on GBS in the Lithuanian population, the results presented are crucial to “better understand global spread of GBS genotypes and antimicrobial resistance patterns” – as the authors summarized their paper. In turn, determination of GBS serotypes distribution may lead to the development of an innovative vaccine that would protect, above all, newborns against life-threatening GBS infections. The methods used in the presented study were appropriate, however some clarification should be considered. I recommend that paper to be accepted for publication due to the importance of the topic, but after a minor essential revision that addresses the following concern. My major comments are listed below.

Introduction: Please briefly describe the genes, which were examined in your study (gyrA, parC, pbp1a, pbp2x)

Authors’ response: We have added the information on the surface proteins as well as mentioned antibiotic resistance genes (gyrA, parC, pbp1a, pbp2x) to the introduction, please see page 2, lines 63-67.

Line 59: Please, rephrase the following expression: “clear need”, please rephrase “circulating antimicrobial resistant phenotypes”

Authors’ response: this sentence has now been rephrased, please see page 2, lines 67-69.

Line 61: In the sentence: “The surveillance of circulating GBS serotypes and genotypes will provide essential information for estimating GBS vaccine efficacy in the future” there is too far-reaching cause-and-effect relationship, please improve the sentence

Authors’ response: this sentence has now been removed.  

Line 77: What was the source of remaining isolates?

Authors’ response: The source for 42 GBS isolates: 35 were isolated from vaginal swab samples, 6 were isolates from urine samples and 1 was isolated from a blood sample. All the source for the individual isolates is also provided in Suppl Table S1, which we have now referenced in the text.

Line 81: Please verify the percentage values. My calculations indicate, that serotype III is present in 33,33% of examined isolates.

Authors’ response: the percentage values have been checked and corrected accordingly.

Lines 84-85: Information of three novel ST’s is duplicated. Please improve it.

Authors’ response: The duplication has been removed.

Line 118: Please rephrase the following sentence: “Six isolates, all of them serotype Ia genotype ST88, did not carry any acquired AMR genes.”

Authors’ response: the sentence has been rephrased.

Line 138: Please rephrase the following sentence: “Figure 2. Previously described novel multidrug resistance MGE ICESag37 found in GBS isolates from Lithuania.” I suggest to avoid “previously”

Authors’ response: We thank the reviewer for this suggestion. We have now decided to move this figure into supplementary material and have rephrased the title as suggested by the reviewer.  

Line 140: use italics in the species name

Authors’ response: the species name has been put in italics.

Line 206: Please verify this sentence: “In our dataset 97.6% (41/42) of GBS isolates had at least one of these four proteins, suggesting that this vaccine could also offer high level of coverage for these isolates “. For my best knowledge only one Alp protein can be present in the individual bacterial strain, so the expression “at least one” may be misleading.

Authors’ response: Apologies for the confusion, yes, we meant that only one of the four proteins targeted by the vaccine was present in 41/42 of our isolates. We have corrected this sentence now, please see page 4, lines 170-172.

Line 208: Please revise: “Notwithstanding”

Authors’ response: we have replaced “Notwithstanding” with “Unfortunately”.

Line 234: Please revise: “Where available anonymised, double-coded data on clinical diagnosis, age of the patient and routine antibiotic susceptibility testing result was collected.”

Authors’ response: we have rephrased this sentence.

Lines 235-237: Please name the examined antibiotics.

Authors’ response: we have added information on the antibiotics used to test for antimicrobial susceptibility, please see page 6, lines 265-266.

Line 245: Please include the name of the producers of mutanolysin and lysozyme

Authors’ response: the manufacturer information has been included for mutanolysin and lysozyme, please see page 7, lines 285-286.  

Line 227: section “Materials and Methods”, please briefly describe how the samples were collected. The following sentence: “Of the 42 GBS isolates, 35 isolates were retrieved from vaginal swab samples, six from 75 urine samples, and one from a blood sample of an individual with confirmed bacteraemia.” As well as “…with the median age of 32y for this cohort” should be moved to “Material and Methods” section. Was each sample collected from individual patient or some samples (e.g. swab and urine) had been collected from the same patient? How bacterial cultivation was carried out?

Authors’ response: We have added information that bacteria were cultivated following CLSI guidelines and local microbiology diagnostic procedures. As far as it is known to us, we did not more than one sample from the same patient. We have updated the Materials and Methods sections accordingly, please see page 6, lines 256-263.

Lines 272-274: Please state what surface proteins were studied. More detailed procedures of the proteins’ detection would improve the paper value. Please improve.

Authors’ response: we have added additional details for detecting surface protein genes to confirm presence or absence of these surface proteins in the GBS dataset. Please see page 7, lines 318-324.  

Line 279: Double space, please improve

Authors’ response: text has been checked for double spaces, some of the formatting is due to the template provided. We hope that these extra spaces will get removed once the manuscript is converted to online version.

Line 291: Please rephrase “Where no characteristic genes were identified…”

Authors’ response: the sentence has been rephrased.

Line 301: Please develop the phrase “GTR+G+ASC”

Authors’ response: the following information has been added “..with “GTR+G+ASC” model (which combines the following models: general time reversible model with unequal rates and unequal base frequency (GTR model); discrete Gamma model with default 4 rate categories (G model) and an ascertainment bias correction model (ASC))”, please see page 8, lines 353-356.